# Biomechanical Efficacy and Effectiveness of Orthodontic Treatment with Transparent Aligners in Mild Crowding Dentition—A Finite Element Analysis

**DOI:** 10.3390/ma15093118

**Published:** 2022-04-26

**Authors:** Jeong-Hee Seo, Min-Seok Kim, Jeong-Hyeon Lee, Emmanuel Eghan-Acquah, Yong-Hoon Jeong, Mi-Hee Hong, Bongju Kim, Sung-Jae Lee

**Affiliations:** 1R&D Center, DENTIS Co., Ltd., Daegu 41065, Korea; sjh00@dentis.co.kr; 2Department of Biomedical Engineering, Inje University, Gimhae 50834, Korea; kms20162296@gmail.com (M.-S.K.); jeonghyeon6089@gmail.com (J.-H.L.); 3School of Health Sciences and Social Work, Griffith University, Gold Coast, QLD 4222, Australia; eghanacquah@gmail.com; 4Osong Medical Innovation Foundation, Cheongju 28160, Korea; yonghoonj186@kbio.kr; 5Department of Orthodontics, School of Dentistry, Kyungpook National University, Daegu 41940, Korea; mhhong1208@knu.ac.kr; 6Dental Life Science Research Institute, Seoul National University Dental Hospital, Seoul 03080, Korea

**Keywords:** orthodontic treatment, transparent aligner, finite element analysis, model activation/de-activation method, biomechanical efficacy and effectiveness

## Abstract

Orthodontic treatment increasingly involves transparent aligners; however, biomechanical analysis of their treatment effects under clinical conditions is lacking. We compared the biomechanical efficacy and effectiveness of orthodontic treatment with transparent aligners and of fixed appliances in simulated clinical orthodontic treatment conditions using orthodontic finite element (FE) models. In the FE analysis, we used Model Activation/De-Activation analysis to validate our method. Fixed appliances and 0.75-mm and 0.5-mm thick transparent aligners were applied to a tooth-alveolar bone FE model with lingually-inclined and axially-rotated central incisors. Compared to the fixed appliance, the 0.75-mm and 0.5-mm transparent aligners induced 5%, 38%, and 28% and 21%, 62%, and 34% less movement of the central incisors and principal stress of the periodontal ligament and of the alveolar bone, respectively, for lingual inclination correction. For axial-rotation correction, these aligners induced 22%, 37%, and 40% and 28%, 67%, and 48% less tooth movement and principal stress of the periodontal ligament and of the alveolar bone, respectively. In conclusion, transparent aligners induced less tooth movement, it is sufficient for orthodontic treatment, but 0.5-mm aligners should be used for only mild corrections. Additionally, the Model Activation/De-Activation analysis method is suitable for FE analysis of orthodontic treatment reflecting clinical treatment conditions.

## 1. Introduction

Malocclusion refers to misalignment of teeth or misalignment between the dental arches of the maxilla and mandible. Malocclusion is largely affected by genetic factors; however, it can also be caused by acquired factors, such as poor dental care and dietary habits in infancy [1,2]. If the occlusion problem persists, it caused distal extension of the mandible, resulting in severe malocclusion due to facial asymmetry [2,3]. Malocclusion patients may undergo orthodontic treatment to achieve proper alignment. This treatment is based on Sandstedt’s pressure-tension theory for tooth movement [1]. According to this theory, external force in the desired direction compresses the periodontal ligament (PDL) between the teeth and alveolar bone in the direction of movement. This activates osteoclasts and leads to resorption of the alveolar bone. In the opposite direction, PDL is tensioned, causing formation of new bone by activation of osteoblasts. Together, this leads to tooth movement, resulting in alignment of teeth.

Orthodontic treatment is conducted using an orthodontic appliance. Generally, a fixed appliance, consisting of a bracket that is attached to the teeth and arch-wires connecting the brackets, is used. The fixed appliance may move the teeth as desired depending on bending of the arch-wire, and provides constant orthodontic force to the teeth. These appliances may be used in all types of orthodontic treatments. However, the brackets attached to the teeth are not aesthetically pleasing to patients, and cause discomfort during mastication. This also leads to difficulties in dental self-care of patients [2,4,5,6]. In particular, constant high orthodontic force causes orthodontic pain and can lead to root resorption after treatment [2,7]. To overcome these limitations, Align Technology (U.S) first developed a removable transparent aligner, Invisalign^®^, made of polymer materials that can be attached and detached as needed by patients themselves. Orthodontic treatment using this transparent aligner has been conducted in many patients [2,5,8,9]. Based on a survey assessing pain in patients undergoing orthodontic treatment using transparent aligners or fixed appliances, orthodontic pain was greater in patients using fixed appliances than in those use transparent aligners. These findings suggested that transparent aligners provide lower force to the teeth than fixed appliances; however, this suggestion was controversial in clinical [10]. 

Orthodontic treatment using transparent aligners requires a longer treatment period, as they yield less tooth movement than fixed appliances [11,12]. This may be attributed to the lower force applied by the polymer materials than by the metal used in fixed appliances [11]. However, clinical studies have difficulty in presenting the results of qualitative analysis of orthodontic force and tooth movement in orthodontic treatment using transparent aligners compared to that using fixed appliances. Instead, biomechanical experiments and finite element (FE) analysis studies are used to obtain such information. Inoue et al. [13] measured orthodontic force in vitro for 14 days after installing transparent aligners and fixed appliances in a dental model with a 3° inclination of the central incisor. Fixed appliances maintained the force with less than 5% change, while the force was decreased by approximately 70% within 2 days with transparent aligners. Moreover, Li et al. [14] compared the force according to the thickness of transparent aligners in an in vitro study. The force was reduced by 50% and 75% on average after 8 h and 4 days, respectively. Although these in vitro findings are useful for analysing the force derived from fixed appliances, the experiments do not fully recapitulate the bone resorption and bone formation effects on PDLs. This is a limitation in analysis of the effects of orthodontic treatment based on the pressure-tension theory.

To compensate for the drawbacks of these clinical and in vitro studies, several studies have evaluated the biomechanical effects of orthodontic treatment through FE analysis. FE analysis is possible to simulate the orthodontic treatment through the reflect of PDLs and to predict phenomena that are difficult to investigate through in vitro experiments [15,16,17,18,19,20]. Gomez et al. [16] and Kim et al. [17] simulated a model of the tooth-alveolar bone structure through a FE method and analysed the biomechanical effects of orthodontic treatment using transparent aligners. Yokoi et al. [18] also used FE analysis to evaluate the behaviour of teeth during orthodontic treatment using transparent aligners. Furthermore, Liu et al. [19] compared the stress in PDL according to the thickness of transparent aligners and simulated wearing of transparent aligners through FE analysis. However, these studies analysed tooth movement and PDL stress by directly applying force to a single tooth [16,17]. Yokoi et al. assumed the alveolar bone as a rigid body and did not fully recapitulate the actual orthodontic treatment conditions [18]. Liu et al. applied transparent aligners to the positions of normally aligned teeth and did not assess the effects of orthodontic treatment using transparent aligners on misaligned teeth [19]. Most studies applied linear material properties to transparent aligners and did not reflect the proper conditions of orthodontic treatment using transparent aligners. In addition, no study directly compared transparent aligners and fixed appliances [16,17,18,19]. In particular, no study that conducted FE analysis reflected the actual orthodontic treatment environment, except the study by Zhou et al. [20]. However, these studies have limitations in that they do not apply actual orthodontic forces or consider the nonlinear material properties of the PDL. Currently there is a lack of literature on a finite element method that can effectively simulate orthodontic treatment while considering accurate material properties of the PDL and the actual treatment environment.

Therefore, in this study, we conducted a FE analysis of orthodontic treatment using fixed appliances and transparent aligners on a tooth-alveolar bone FE model that consisted of alveolar bone, PDLs, and multiple teeth, which had not been reflected in previous studies. In particular, the FE analysis was performed by applying the actual orthodontic force and assigning the nonlinear properties of PDLs similar to the actual orthodontic treatment situation. We compared the in vitro orthodontic force-measurement test and FE analysis to assess whether the loading condition reflected the conditions of orthodontic treatment. Using the verified loading conditions, we compared the biomechanical effects and efficacy of orthodontic treatment using transparent aligners and fixed appliances.

## 2. Materials and Methods

### 2.1. Establishing an In Vitro Orthodontic Force-Measurement Test Environment for Finite Element Model Development and Verification of Loading Conditions

#### 2.1.1. Finite Element Model Development

A three-dimensional (3D) normal tooth-alveolar bone FE model, consisting of teeth, PDLs, cortical bone, cancellous bone, and gingiva, which has been previously described [21], was used in this study (Figure 1). Briefly, the FE model consisted of teeth and mandibular alveolar bone from the central incisor (#31) to the second premolar (#35), and was established using cone-beam computed tomography (CBCT) data of a 27-year-old healthy woman with normal occlusion, proper alignment, and healthy teeth, who did not have periodontitis and had not undergone orthodontic treatment. Based on previous studies, linear properties were assumed for the teeth, alveolar bone, and gingiva. Non-linear properties were adopted for PDL to reflect clinical orthodontic treatment conditions accurately (Table 1, Figure 2). Components assigned linear material properties were meshed with four-node tetrahedral (C3D4) elements. PDL and aligner components, which were assigned nonlinear material properties, were meshed with four-node linear tetrahedrons (C3D4H) and 20-node quadratic (C3D20H) element types, respectively.

We used a previously published model of a transparent aligner (Figure 3) [21]. The transparent aligners had a thickness of 0.5 and 0.75 mm, which are the common thicknesses of aligners used in clinical treatment. The shape of the crown portion, with the tooth exposed outside the gingiva and covering part of the gingiva in the normal tooth-alveolar bone model was offset by the thickness of the transparent aligners to create a FE model. To reflect the conditions of transparent aligner orthodontic treatment in FE analysis, the material properties of the transparent aligners were set according to the mechanical test method stipulated by the Korean Ministry of Food and Drug Safety and ISO. The aligners were manufactured using 0.5-mm- and 0.75-mm-thick Duran CA^®^ transparent aligner manufacturing sheets (Scheu-Dental GmbH, Iserlohn, Germany). The mechanical test was then conducted, and the stress-strain results were calculated and applied to the model (Figure 2). A 3D FE model of a fixed appliance was created by reverse-engineering a standard-shape bracket (Archist Edgewise Standard^®^, DAESEUNG Co., Ltd., Seoul, Korea) and 0.016” round-type arch-wire (SS-flex arch wire^®^, DAESEUNG Co., Ltd., Seoul, Korea) used in clinical practice (Figure 3). A model scanner (D700, 3Shape Inc., Copenhagen, Denmark) was used to acquire data on the outer shape of the fixed appliance bracket, and the data were reverse-engineered using 3D design software (SolidWorks 2019, DSS Simulia, Providence, RI, USA) to create FE models. For the ligature, the space between the brackets was set as a line element to prevent separation of the arch-wire from the bracket during FE analysis. The material properties of the fixed appliance were as follows: the bracket and arch-wire had an elastic modulus of 200 GPa and a Poisson’s ratio of 0.3 of SUS 304L, provided by the manufacturer (DAESEUNG Co., Ltd., Seoul, Korea) (Table 1).

We used the same malocclusion tooth-alveolar bone model that we used in our previous study [21]. The central incisor (#31) position was modified in the present model. The central incisor had an inclination of 1.5° in the lingual direction with respect to the center of resistance of the tooth in the sagittal plane, and an axial rotation of 5° from the distal to the mesial direction with respect to the tooth axis in the transverse plane [21]. These parameters were chosen to create malocclusion models that required correction of inclination and axial rotation of the central incisor (Figure 4). The maximum tooth deformation value suggested by the National Health Service (UK) and Align Technology was used [23,24]. The treatment plan, devised by an orthodontist, was to move the deformed central incisor to its normal position. All crowding tooth deformations were simulated using 3D model-based orthodontic treatment planning software (DICAON-4D, DENTIS Co., Ltd., Daegu, Korea). The material properties of the malocclusion models were the same as those of the normal model. Six models were created: malocclusion models of inclination and axial rotation for 0.5- and 0.75-mm-thick transparent aligners and for a fixed appliance (Figure 5).

An additional FE models was developed to verify the validity of the loading conditions used to reflect the orthodontic treatment conditions (Figure 6). Data of seven teeth from the left mandible (#31–#37) and alveolar bone were acquired from CBCT data of healthy individuals that were used to develop the normal tooth-alveolar bone model. A full-arch tooth model of the mandible was designed by symmetrically aligning the data of the left and right sides. The transparent aligner was simulated in a shape suitable for a dental model of the entire mandible, as described above. The arch-wire of the fixed appliance was modelled according to the placement of the teeth on the bracket model used for each tooth. The material properties of the transparent aligners and fixed appliance were equivalent to those of the model used for orthodontic treatment analysis. To recapitulate the in vitro environment, the dental model had the same material properties as the photo-polymerized 3D printed dental model.

#### 2.1.2. Establishment of In-Vitro Orthodontic Force-Measurement Test Environment

To verify the validity of the simulated fixed appliance finite element model and force conditions for reflecting clinical orthodontic treatment conditions, in vitro orthodontic force was measured (Figure 7) [20,24]. As described by Zhou et al. [20], an experimental environment was established to measure the orthodontic force generated by the transparent aligners and the fixed appliance. The orthodontic force on the central incisor (#31) was measured as in orthodontic treatment analysis. To measure orthodontic force, a loadcell was mounted on the central incisor. However, orthodontic force cannot be accurately measured when teeth are attached to the alveolar bone [20]. Therefore, in the 3D shape used for the verification orthodontic model, the central incisor was deleted, and the central incisor on which the loadcell was mounted was designed separately. In addition, a test jig was designed by integrating the orthodontic model with a frame used for placement of the loadcell. The central incisor for measurement of the orthodontic force was inclined 5° to the buccal direction, and after modelling for placing the loadcell, interference with the test jig was assessed. After designing the test jig, the orthodontic model and test jig for manufacturing the transparent aligners and the central incisor for force measurement were printed using a photo-polymerized 3D-printer (ZENITH D, DENTIS Co., Ltd.) and photopolymer resin for dental model printing (ZMB-1000 Model, DENTIS Co., Ltd.). After printing, the loadcell (DAQ Nano 17/6Axis, National Instruments Corp, Austin, TX, USA) was placed on the central incisor, which was then placed on a test jig.

Transparent aligners for measurement of orthodontic force were manufactured using the same method as in the clinical production of sheet material used for model development. The transparent aligners were inserted into thermoforming equipment (Ministar S, Scheu-Dental GmbH, Iserlohn, Germany) with 3D-printed orthodontic models. Five transparent aligners were manufactured for each of the 0.5- and 0.75-mm thicknesses. 

The same fixed appliance used in model development was tested in this part of the study. The brackets were attached to the test jig and central incisor using Andrew’s guide method [25]. Arch-wires were connected to each bracket and were fixed using a ligature clip. Five test jigs were prepared for measurements of the fixed appliance.

### 2.2. Verification of Loading Conditions

#### 2.2.1. Finite Element Anlysis and In-Vitro Orthodontic Force-Measurement Test for Verification of Loading Conditions

Both transparent aligners and fixed appliances are manufactured with consideration of the location after teeth movement, rather than before this movement, to move teeth into their required position. In orthodontic treatment, arch-wires of fixed appliance and transparent aligners are deformed and settled on the position before tooth movement. After binding, orthodontic treatment is performed using the inherent property of arch-wires and transparent aligners of tending to return to their original shape. This condition is difficult to recapitulate using general force and contact conditions in FE analysis software. Therefore, in this study, a “Model Activation/De-Activation” method, available among different interaction functions of ABAQUS, a commercial FE analysis software, was used [26]. This method is used to predict the effects of deformation of one structure on another structure with different material properties or shapes. For example, in an analysis to predict spring elasticity, the analysis is completed with the spring pressed under general loading and boundary conditions. Therefore, the restoration characteristics of the spring cannot be predicted. As shown in Figure 8, when we prepared the model from the primary analysis with a blue dash box and red line for pressuring and activating the prepared model for pressure, we obtained the stress from deformation of the spring. After the primary analysis, deactivation of the blue dash box and red line and activation of the spring allowed restoration of the deformed spring through stress-relaxation effects [26]. Based on this analysis method, we used the “Model Activation/De-Activation” loading conditions, which reflect orthodontic treatment conditions.

The loading condition was set to de-activate the orthodontic model for verification, to avoid affecting the analysis. The simulated central incisor (which was set as a rigid body, in blue in Figure 9) of the transparent aligners and fixed appliance with normal occlusion was moved to the position of the deformed central incisors of the orthodontic model for the verification analysis. Deformation of the fixed appliance that is in contact with the dental model causes stress. Once the movement of teeth is complete, the tooth model is de-activated, and the orthodontic model is activated, resulting in activation of the contact condition between the orthodontic appliance and orthodontic model to verify fixation of the transparent aligners and fixed appliance to the teeth before orthodontic treatment.

The second stage was then analysed. The spring-back property of the arch-wire of the transparent aligners and the fixed appliance, which allows it to return to its original shape, is used to perform orthodontic treatment without additional loading conditions. The second stage was analysed once the orthodontic appliance was fixed, and the analysis was completed when the orthodontic appliance could not return to its original shape and orthodontic treatment ended (Figure 9) [20]. These Model Activation/De-Activation loading conditions were also applied to the orthodontic treatment analysis.

The tooth alignment of the orthodontic model for verification was as follows. The central incisor used to measure orthodontic force was inclined 5° in the buccal direction, as compared to the central incisor of normal teeth. All teeth other than the central incisor were in the normal positions, with no contact conditions. Both transparent aligners and fixed appliances were set in contact with the orthodontic model used for verification and the normal tooth alignment model. As previously described [21], the transparent aligners were frictionless. The bracket and teeth were set to be completely fused. The bracket and arch-wire were frictionless. To prevent separation of the arch-wire from the bracket during orthodontic treatment analysis, a line element was added to the gap between the brackets. To reflect the same conditions as in the in vitro experiment, the boundary condition for the verification analysis was set at 6 degrees of freedom for the lower surface (basal plane) of the orthodontic model (Figure 10).

In the verification analysis, the contact pressure and contact area of the central incisor of the orthodontic model used for verification were multiplied to calculate the orthodontic force. The orthodontic force of each orthodontic appliance was measured for the start and end of the second stage.

#### 2.2.2. In-Vitro Orthodontic Force-Measurement Test

The orthodontic force was measured for 24 h for each orthodontic appliance, as previously described [21,24]. The force was measured by connecting orthodontic force measurement software (Data Acquisition Tool, National Instruments) to the loadcell. For the initial 10 min, the orthodontic force was measured at 10 Hz. For the next 23 h and 50 min, data were measured at 1 Hz. The orthodontic force measurements were compared to those of the verification analysis, to confirm the validity of the developed finite element model and loading conditions.

### 2.3. Orthodontic Treatment Finite Element Analysis

To recapitulate the clinical orthodontic treatment conditions, orthodontic treatment analysis was conducted using the Model Activation/De-Activation method (Figure 11). To analyse orthodontic treatment, a rigid-body model of the teeth, transparent aligners, and fixed appliance, according to the planned orthodontic treatment, were used.

The rigid-body teeth model had normal occlusion, and the orthodontic tooth-alveolar bone model required correction of lingual inclination and axial rotation of the central incisor for orthodontic treatment. Analysis was conducted in two stages, as in the verification analysis. In the first stage, the central incisor of the rigid-body teeth model was forcefully moved to deform the arch-wire and transparent aligners, and then the orthodontic appliances were in (contact with) the orthodontic tooth-alveolar bone model. In the second stage, orthodontic treatment of the central incisor, using the spring-back characteristic of the orthodontic appliance, was analysed. The contact conditions were the same as those in the verification analysis, and both sides of the orthodontic tooth-alveolar bone were constrained in all directions (Figure 12).

After analysis of orthodontic treatment, movement of the central incisor and principal stress of the PDL and of the alveolar bone were compared for the transparent aligners and fixed appliance. Additionally, the peak Von Mises stress (PVMS) of transparent aligners was analysed. Movement of teeth by transparent aligners and the fixed appliance was compared to investigate the effectiveness of orthodontic treatment. Moreover, the principal stress on the PDL and alveolar bone by transparent aligners and the fixed appliance was compared to predict the likelihood of alveolar bone remodelling after teeth movement. The PVMS of transparent aligners was compared to the yield and maximum stresses of the material to analyse the possibility of deformation and breakage of the transparent aligners. In the analysis of the effectiveness and efficacy of orthodontic treatment using transparent aligners, application of the attachments that are used in clinical practice was not considered.

All FE models were developed and analysed using commercial FE software (ABAQUS/CAE v2018, DSS Simulia, Providence, RI, USA).

## 3. Results

### 3.1. Verification of Loading Conditions

To verify the applicability and validity of the Model Activation/De-Activation loading conditions for orthodontic treatment analysis, in vitro orthodontic force was measured, and FE analysis was conducted under identical conditions.

As illustrated in Figure 13, 0.75-mm-thick and 0.5-mm-thick transparent aligners, and the fixed appliance initially had 0.2 N, 0.1 N, and 0.9 N greater orthodontic force, respectively, than did the test models. Finally, there was a difference of 0.3 N, 0.3 N, and 0.8 N in orthodontic force between FE analysis and test 0.75-mm-thick and 0.5-mm-thick transparent aligners, and fixed appliance, respectively. This suggested that the Model Activation/De-Activation loading condition applied in this study was applicable for FE analysis of orthodontic treatment and reflected clinical orthodontic treatment conditions.

### 3.2. Orthodontic Treatment Results of Orthodontic Appliances

#### 3.2.1. Central Incisor Movement

Movement of central incisors by the 0.75-mm- and 0.5-mm-thick transparent aligners was compared to that by the fixed appliance (Figure 14). 

In orthodontic treatment for a 1.5° lingual inclination, the 0.75-mm and 0.5-mm transparent aligners rotated the central incisor by 1.42° and 1.19°, respectively. On the other hand, the fixed appliance rotated the central incisor by 1.51°. Compared to the fixed appliance, the 0.75-mm and 0.5-mm transparent aligners were 94% and 79% effective, respectively, in inducing tooth movement. 

In orthodontic treatment for a 5° axial rotation, the 0.75-mm and 0.5-mm transparent aligners rotated the central incisor by 3.79° and 3.51°, respectively, while the fixed appliance rotated the central incisor by 4.83°. Thus, compared to the fixed appliance, the 0.75-mm and 0.5-mm transparent aligners were 78% and 72% effective, respectively, in inducing tooth movement.

#### 3.2.2. Principal Stress of the Periodontal Ligament and Alveolar Bone

We compared orthodontic treatment for lingual inclination and axial rotation of central incisors by comparing the principal stresses of the PDL and alveolar bone produced by the transparent aligners with that produced by the fixed appliance (Figure 15 and Figure 16).

The principal stress of the PDL was greater in treatment of axial rotation than in treatment of lingual inclination for the two transparent aligners. In treatment of lingual inclination, the 0.75-mm transparent aligner led to 31% tension and 45% compression, and the 0.5-mm transparent aligner led to 26% tension and 31% compression, as compared to the fixed appliance. For treating axial rotation, the 0.75-mm transparent aligner caused approximately 46% tension and 28% compression, and the 0.5-mm transparent aligner caused 24% tension and 22% compression, as compared to the fixed appliance.

Unlike the principal stress of the PDL, the principal stress of the alveolar bone was higher in lingual inclination treatment than in axial rotation treatment. Compared to the fixed appliance, treatment of lingual inclination with the 0.75-mm transparent aligner caused 76% tension and 67% compression, and the 0.5-mm transparent aligner induced 72% tension and 61% compression. For axial rotation treatment, compared to the fixed appliance, the 0.75-mm transparent aligner led to 66% tension and 54% compression, and the 0.5-mm transparent aligner led to 57% tension and 46% compression.

In orthodontic treatment of lingual inclination, the 0.75- and 0.5-mm transparent aligner and fixed appliance induced the greatest tension stress at the lingual-cervical side of the central incisor’s PDL. The greatest compression stress was observed in the buccal-cervical direction of this ligament. Both 0.75- and 0.5-mm transparent aligners induced the most tension in the buccal-cervical direction and the most compression in the buccal-apical direction of the alveolar bone. On the other hand, the fixed appliance induced the maximum principal stress in the same areas as in PDL.

In correction of axial rotation, the tension and compression principal stresses were induced in the same areas of the PDL and alveolar bone by all orthodontic appliances. Most tension was observed in the lingual-mesial direction, and compression was observed in the buccal-mesial direction of the PDL and alveolar bone (Figure 17).

#### 3.2.3. Peak Von-Mises Stress of Transparent Aligners

After orthodontic treatment analysis, the PVMS of the transparent aligners was analysed at the time of insertion, when this value was highest. In both inclination and axial rotation correction, the 0.75-mm and 0.5-mm transparent aligners caused greater stress than the yield stress of the transparent aligner material; however, the stress was lower than the ultimate stress. In treatment of lingual inclination, the 0.75-mm and 0.5-mm transparent aligners showed 16% and 14% increased stress, respectively, as compared to the yield stress. 

In correction of axial rotation, the 0.75-mm and 0.5-mm transparent aligners increased stress by 37% and 34%, respectively, relative to yield stress. The PVMS of the transparent aligners was observed at points where the aligners contacted the deformed area of each tooth (Figure 18).

## 4. Discussion

As in vitro experiments cannot recapitulate the pressure-tension theory related to teeth, PDL, and the alveolar bone, on which orthodontic treatment is based, analysis of the effects and utility of orthodontic treatment using transparent aligners has been limited. On the other hand, FE analysis allows assessment of the biomechanical effects of orthodontic treatment using transparent aligners. FE analysis is a repeatable method of research, and is known as an efficient method to investigate dental biomechanics as well as orthopaedics [14,15,16,17,18,19,20,21,27,28,29]. Nevertheless, such FE analysis studies have to date not accurately reflected clinical orthodontic treatment conditions. Force was directly applied to teeth without an orthodontic appliance [16,17], and simplified FE models were used [18]. To address these shortcomings, we here simulated the clinical conditions of orthodontic treatment through FE analysis to evaluate the biomechanical efficacy and effectiveness of orthodontic treatment with transparent aligners as compared to that with fixed appliance. We proved that the Model Activation/De-Activation analysis method is suitable for FE analysis of orthodontic treatment and reflects clinical treatment conditions. Moreover, we found that, although transparent aligners induced less tooth movement, it is sufficient for orthodontic treatment. Nevertheless, 0.5-mm-thick aligners, rather than 0.75-mm-thick aligners, should be used for mild corrections.

We used the Model Activation/De-Activation analysis method to reflect the clinical conditions of orthodontic treatment in our analysis. We established an in vitro environment for measuring orthodontic force to confirm the validity of our analysis method. The force applied to the teeth for each orthodontic appliance was measured, and FE analysis was performed under the same conditions as used in the in vitro test environment [20,24]. The difference between the in vitro test and FE analysis environment was less than 3% for transparent aligners and approximately 10% for the fixed appliance, suggesting that the Model Activation/De-Activation analysis method was suitable for simulation of the orthodontic treatment condition. In our previous study [21], a sub-routine analysis method was used. In the first stage, we induced pre-stress to the transparent aligners, and the results of this stage were used as the analysis condition for the second stage of orthodontic treatment analysis. A deformed transparent aligner from the first stage and a non-deformed transparent aligner from the second stage were forced to match. In this method, the analysis could not be conducted if the shape of the two transparent aligner models from the two stages did not match properly [19,21]. Therefore, in this study, FE analysis of orthodontic treatment with the fixed appliance was simulated using the Model Activation/De-Activation function of the software for FE analysis, as described by Zhou et al. [20]. We applied the clinical conditions of treatment using transparent aligners and fixed appliance [26]. We observed that the method could be applied to both transparent aligners and fixed appliances. Unlike the subroutine method, analysis could be performed without forceful matching of the shape of the orthodontic appliance before and after deformation. Thus, Model Activation/De-Activation method may be used for future studies on FE analysis of various orthodontic treatments.

Tooth movement is an important criterion for determining the success of orthodontic treatment [27,28]. Transparent aligners must transmit orthodontic force to the teeth without damage or serious deformation during orthodontic treatment [22,30]. In this study, we observed that 0.75-mm transparent aligners achieved 95% lingual inclination rotation and 76% axial rotation, while 0.5-mm transparent aligners induced 79% lingual inclination rotation and 70% axial rotation, relative to that achieved by a fixed appliance. The increased effects of the 0.75-mm transparent aligner may be because these aligners exerted greater force than the 0.5-mm aligners, as seen in the in vitro orthodontic force analysis. However, in axial rotation, transparent aligners can only transmit force to the crown region. This may cause transparent aligners to exert lower force for axial rotation correction than fixed appliances. In terms of the PVMS of the transparent aligners, axial rotation was improved by 15% as compared to inclination for both 0.75-mm and 0.5-mm transparent aligners. However, 19% less tooth movement was achieved in axial rotation than in inclination correction. This suggest that transparent aligners do not transmit sufficient force for treatment of axial rotation, and that axial rotation correction using transparent aligners may require additional orthodontic treatment. The PVMS of all orthodontic treatments for both 0.75-mm and 0.5-mm transparent aligners was higher than the yield stress and lower than the ultimate stress. The PVMS was higher than the yield stress for teeth requiring orthodontic treatment from the time of transparent aligner insertion. This suggests that the aligner may be deformed from this time-point. Our results show that transparent aligners may be used for orthodontic treatment, even though they led to less tooth movement than fixed appliances, although insufficient tooth movement must be improved in terms of correction of axial rotation.

Changes in PDL due to external force is an important factor in tooth movement induced by orthodontic treatment [2,7,31,32]. In particular, transparent aligners induce tooth movement by continuous application of less force than that applied by fixed appliances [8,9]. Transparent aligners created less tension and compression for both lingual inclination and axial rotation deformation than did the fixed appliance. This was consistent with the tooth movement and PVMS of transparent aligners. Compared to the findings of Bergomi et al. [7] on changes in PDL according to stress-deformation, transparent aligners of all thicknesses led to PDL deformation within an acceptable range. Based on these findings, transparent aligners may be able to transmit adequate force to PDL for orthodontic treatment. However, transparent aligners induced less principal stress than the fixed appliance, which would be associated with a lower rate of alveolar bone reformation during tooth movement, increasing the duration of orthodontic treatment. This finding may be related to a study by Chisari et al. [11] who reported that tooth movement with transparent aligners, according to installation time, was insufficient as compared to that of fixed appliances.

This study had several limitations. First, only unidirectional, mild-crowding dentition that could be corrected using transparent aligners was evaluated. In addition, the fixation attachments that are used for effective tooth movement by transparent aligners were not considered. On the other hand, since this study was no consideration according to the age of orthodontic patients or condition of PDLs, additional research is needed [33]. Future studies should analyse orthodontic treatment using transparent aligners with attachments and should conduct FE analysis of treatment of severe-crowding dentition, with two or more teeth and bidirectional deformation, for a more comprehensive understanding of the biomechanical effects of transparent aligners. Additionally, we will compare the orthodontic force and deformation of orthodontic appliances with results obtained using FE analysis.

## 5. Conclusions

In this study, we simulated the clinical conditions of orthodontic treatment through FEA to evaluate the biomechanical efficacy and effects of orthodontic treatment with transparent aligners. The choice of Model Activation/De-Activation finite element method employed in this study was proven to be suitable for the prediction of orthodontic treatment using transparent aligners and fixed appliances alike. This method is thus recommended for future simulation of orthodontic treatments of different malocclusions. 

The findings of this study showed that in orthodontic treatment using transparent aligners, the principal stresses of the PDL were within the range to induce tooth movement suggesting that transparent aligners are effective for orthodontic treatment. Although the results of this study suggested that transparent aligners induce less tooth movement than the fixed appliance, the effects seemed sufficient for orthodontic treatment. However, in cases involving the correction of axial rotation, additional orthodontic treatment would be necessary. 

## Figures and Tables

**Figure 1 materials-15-03118-f001:**
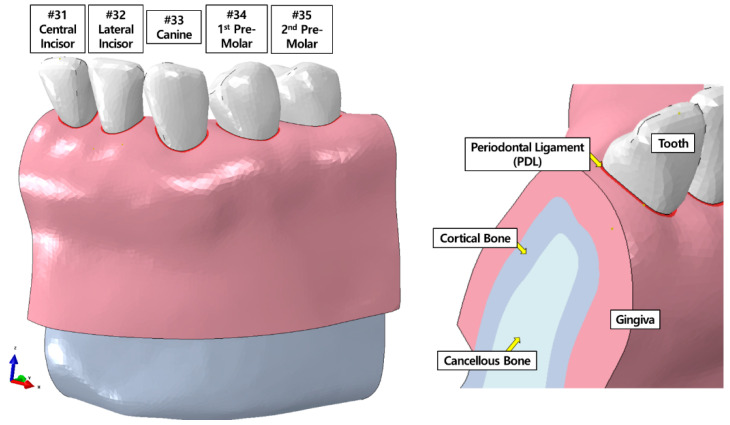
Normal tooth-alveolar bone FE model verified in a previous study [21].

**Figure 2 materials-15-03118-f002:**
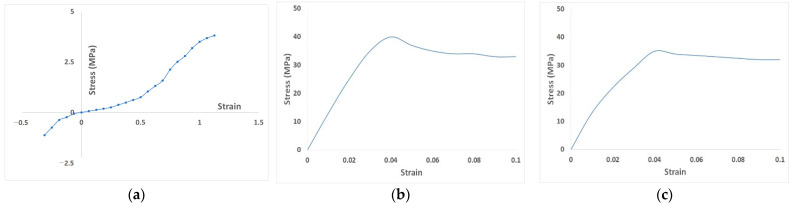
Stress-strain curve for nonlinear material properties of (**a**) periodontal ligaments (PDLs) [21], (**b**) 0.5-mm thick, and (**c**) 0.75-mm thick transparent aligner obtained through testing [22].

**Figure 3 materials-15-03118-f003:**
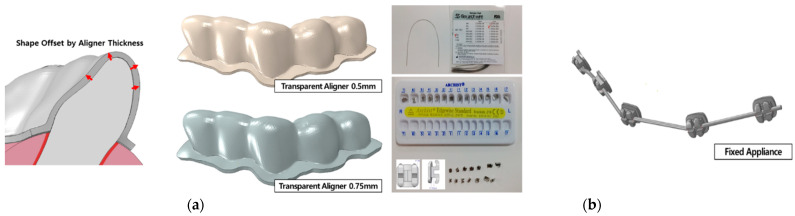
Orthodontic appliance finite element models. (**a**) Model developed by offsetting the outer shape of the crown and gingiva with 0.5-mm and 0.75-mm-thick transparent aligners; (**b**) Model developed through reverse engineering of previously commercialized edgewise standard bracket and arch-wire.

**Figure 4 materials-15-03118-f004:**
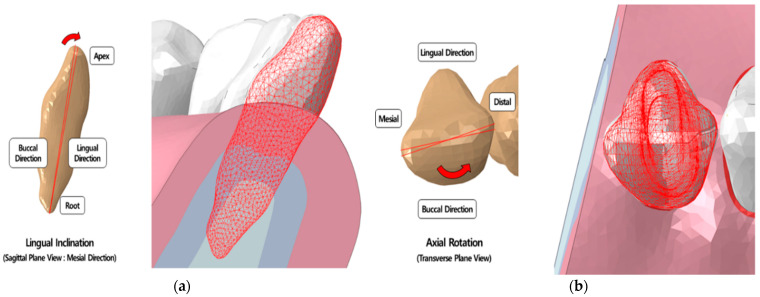
Tooth-alveolar bone model of mild crowding dentition (**a**) Buccal 1.5° inclined rotation of the central incisor (**left**) and deformation of the central incisor compared to normal tooth-alveolar bond model (red); (**b**) Five-degree external axial rotation of the central incisor (**left**) and deformation of the central incisor compared to normal tooth-alveolar bone model (red); the components are equal to that of normal tooth-alveolar bone model except for the central incisor.

**Figure 5 materials-15-03118-f005:**
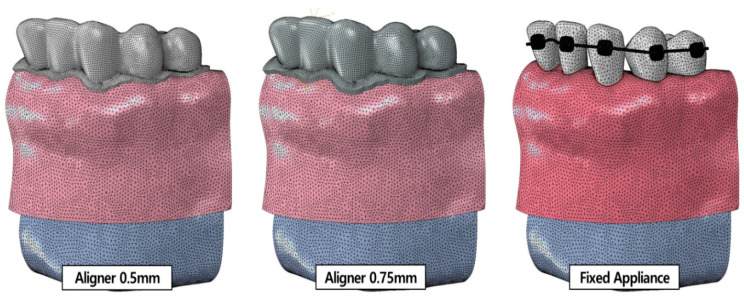
Orthodontic treatment finite element models. (**left**) 0.5-mm and (**center**) 0.75-mm-thick transparent aligners; (**right**) tooth-alveolar model of mild crowding dentition with a fixed appliance.

**Figure 6 materials-15-03118-f006:**
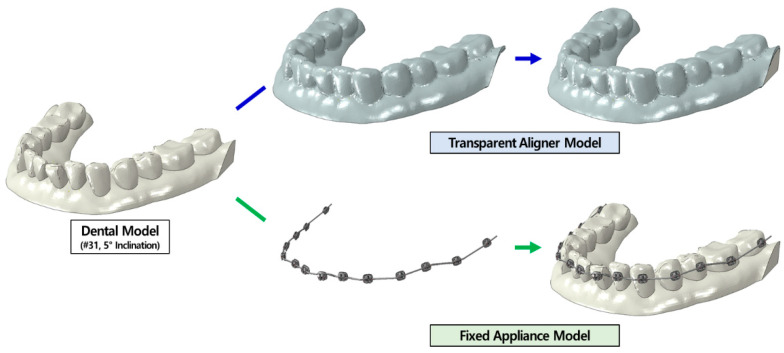
Reconstruction of the finite element models for verification of loading condition. (**left**) Cone-beam computed tomography data for normal tooth-alveolar bone model was used to simulate all mandibular teeth and alveolar bone, and the #31 central incisor for force measurement was inclined at 5° to develop a verification model. (**center**) Development of transparent aligner and fixed appliance models using the same method as the orthodontic appliance model development method. (**right**) Transparent aligner and fixed appliance models were applied to the verification model to simulate the loading condition.

**Figure 7 materials-15-03118-f007:**
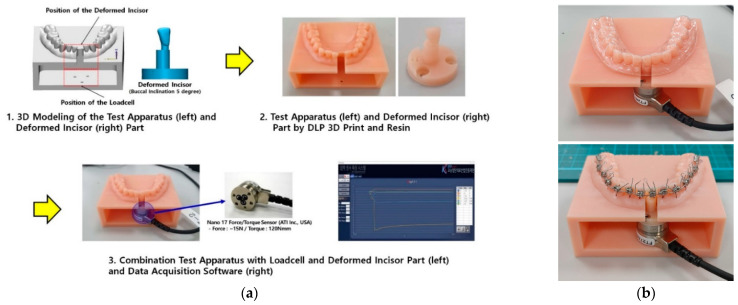
In-vitro orthodontic force measurement environment establishment and testing. (**a**) (1), The central incisor for evaluation of orthodontic force was removed from the 3D-shape data used in the verification orthodontic model. The central incisor mounted on the loadcell was inclined at 5° in the buccal direction. A frame that could hold the loadcell was integrated with the orthodontic model to design the test jig. (2), Thereafter, the test jig was printed using a photo-cured 3D-printer and photo-cured resin material for printing of dental models. (3), After printing, the loadcell and central incisor were combined and placed onto a test jig. The loadcell was connected to the orthodontic force measurement software for data acquisition. (**b**) Data were recorded for 24 h per specimen for each orthodontic appliance. Data were acquired at a rate of 10 Hz for the first 10 min. Thereafter, data were acquired at a rate of 1 Hz for 23 h and 50 min.

**Figure 8 materials-15-03118-f008:**
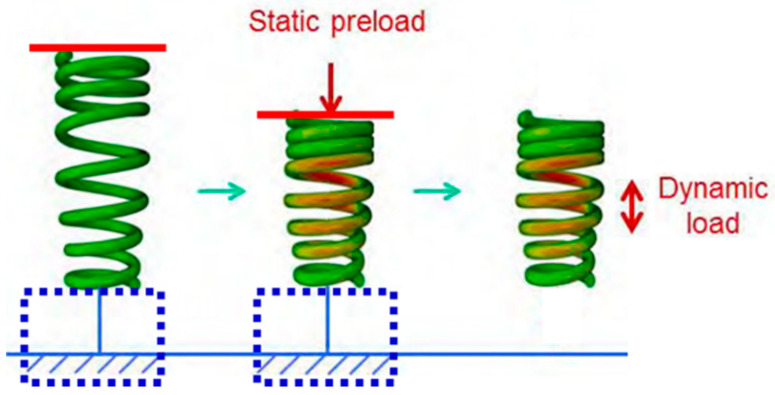
Analysis of spring properties using the Model Activation/De-Activation method [26]. In force/boundary conditions of general finite element analysis, the analysis ends when the spring is compressed, as shown in the middle figure. When the area of force applied (red-line) and the boundary (blue dash box) are de-activated after analysis, the actual spring characteristics, in which the string stressed by deformation returns to its original shape, may be predicted.

**Figure 9 materials-15-03118-f009:**
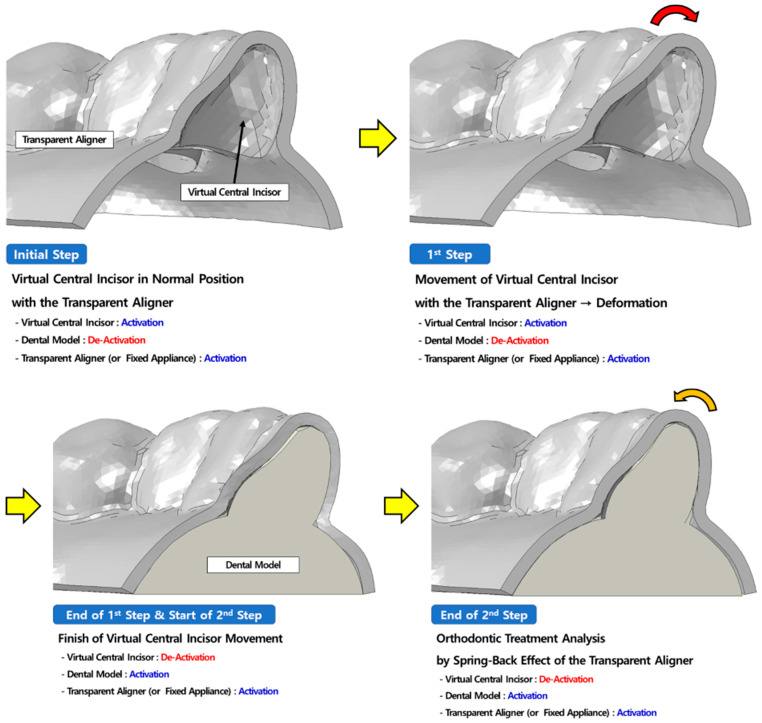
Process of loading condition verification analysis. Initial step: the verification orthodontic model that corresponds to the pre-orthodontic condition is de-activated. The transparent aligner and virtual tooth in normal positions are activated. First step: the virtual tooth is forcefully moved to the pre-orthodontic position, and the transparent aligner is deformed to be worn in the pre-orthodontic state. End of first step and start of second step: After analysis of the first step, the virtual tooth is de-activated, and the dental model is activated (contact between the dental model and transparent aligner is applied). End of second step: Although there is no additional force, deformation of the transparent aligner in the first step applies orthodontic force to the tooth through a stress-relaxation effect to return to its original shape. The fixed appliance was analysed using the same method.

**Figure 10 materials-15-03118-f010:**
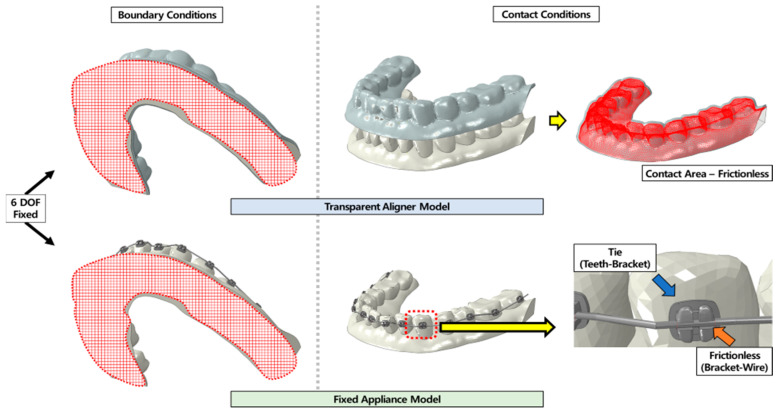
Boundary and contact conditions of loading condition verification analysis. The entire lower surface (red area) of both the fixed appliance and the transparent aligner models were constrained to 6 degrees of freedom to set the boundary conditions. The contact condition was frictionless between the transparent aligner and teeth. In the fixed appliance model, a tie condition was applied for the teeth and bracket, while the bracket and arch-wire contact were considered to be frictionless.

**Figure 11 materials-15-03118-f011:**
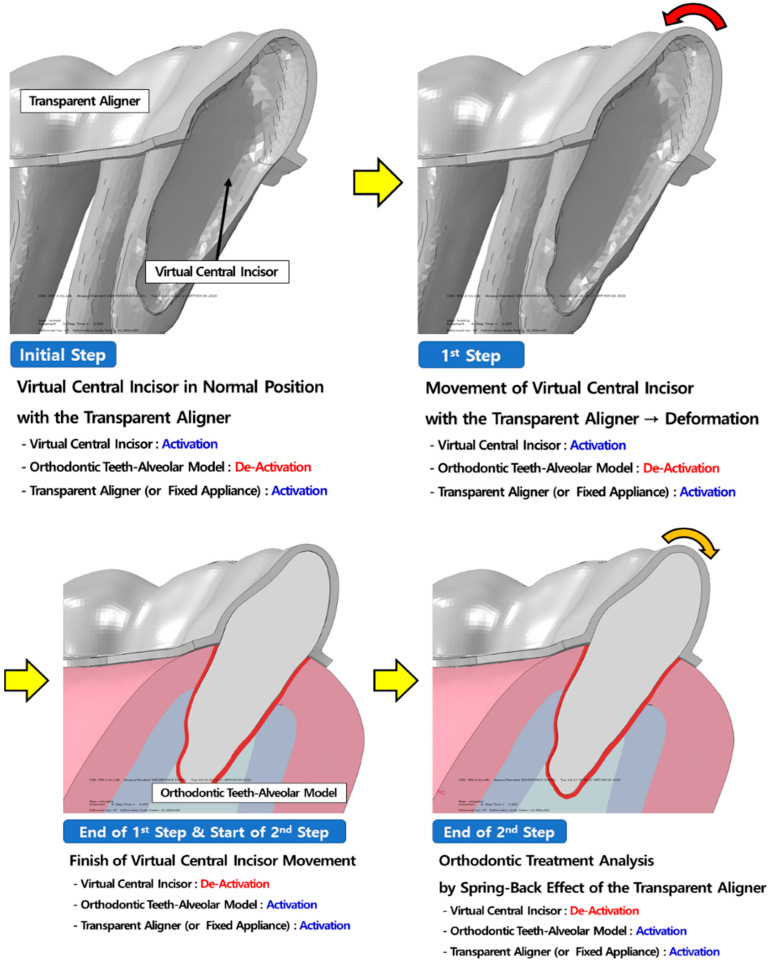
Process of orthodontic treatment analysis. The same method used for verification analysis was used, and the same loading conditions were applied to the analysis of the fixed appliance.

**Figure 12 materials-15-03118-f012:**
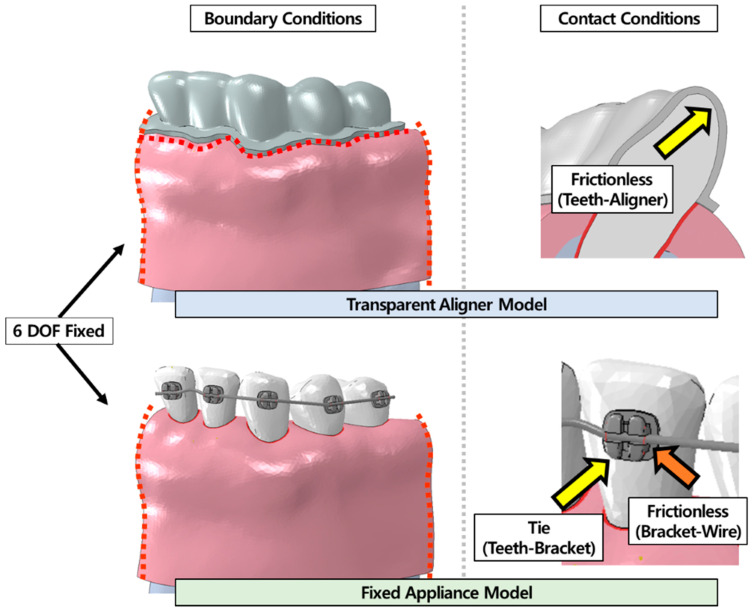
Boundary and contact conditions in orthodontic treatment analysis. On both sides (red dashed line) of the tooth-alveolar bone model with a transparent aligner and fixed appliance constrained to six degrees of freedom. In the transparent aligner model, the bottom surface was set so that there was no movement. The contact condition was frictionless between the teeth and aligner, as in the verification analysis. In the fixed appliance model, a tie condition was applied for the teeth and bracket, while the bracket and arch-wire contact was considered frictionless.

**Figure 13 materials-15-03118-f013:**
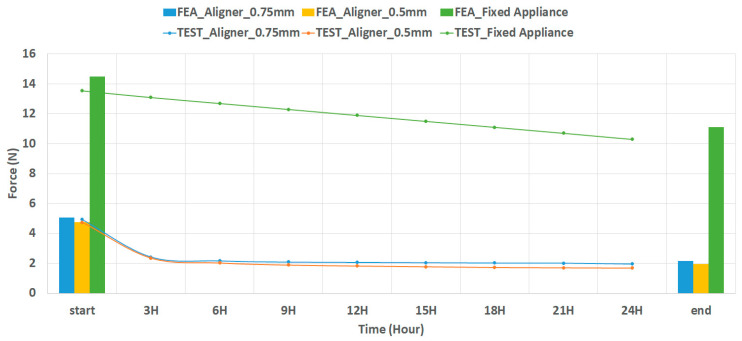
Comparison of the results of loading condition verification analysis and in vitro orthodontic force measurement.

**Figure 14 materials-15-03118-f014:**
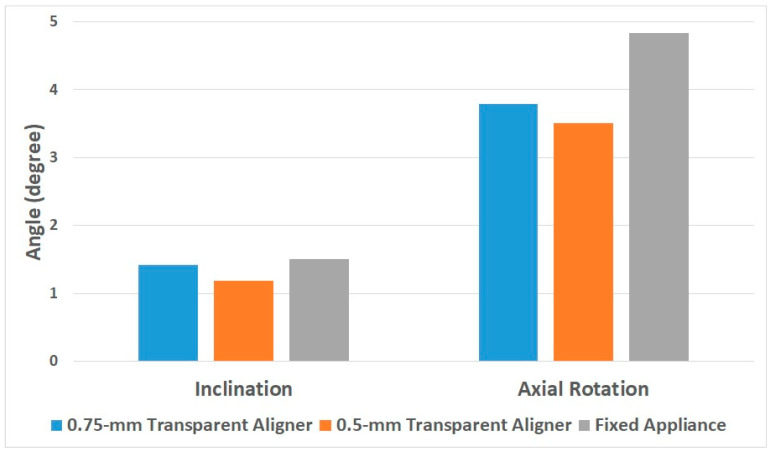
Movement of central incisor in the finite elements analysis of orthodontic treatment.

**Figure 15 materials-15-03118-f015:**
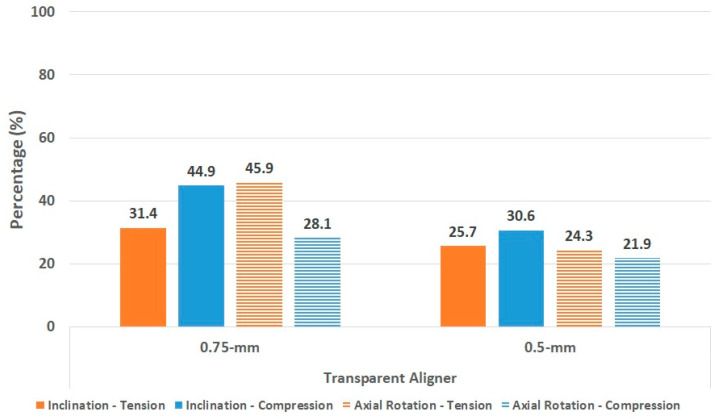
Principle stress of the periodontal ligament induced by the transparent aligner as compared to that induced by the fixed appliance.

**Figure 16 materials-15-03118-f016:**
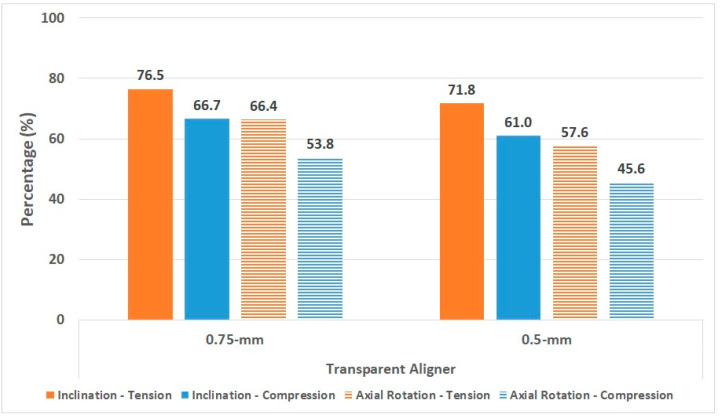
Principle stress of alveolar bone induced by the transparent aligner as compared to that induced by the fixed appliance.

**Figure 17 materials-15-03118-f017:**
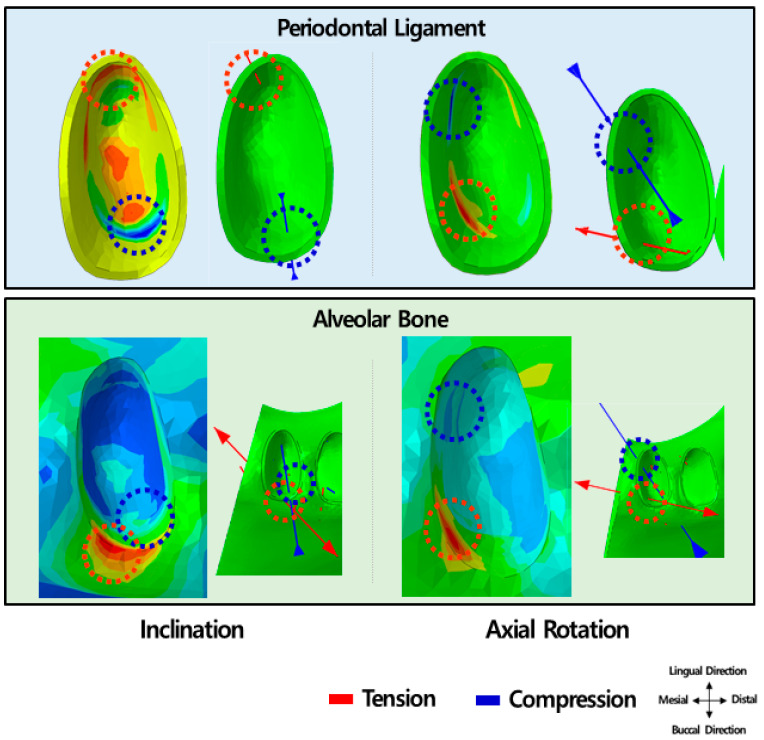
Distribution and direction of principal stress of the periodontal ligament and alveolar bone in orthodontic treatment using a transparent aligner. In the axial rotation treatment, the expected direction and location of the tension and compression principal stress was the same in the periodontal ligament and in the alveolar bone. In inclination treatment, compression was predicted to be in the buccal direction at the middle point between the periodontal ligament and the alveolar bone. In contrast, tension was predicted to be present in the lingual-cervical part of the periodontal ligament and buccal-cervical part of the alveolar bone. Based on pressure-tension theory, in such cases, new bone formation and bone resorption in axial rotation treatment using transparent aligner would proceed without difficulty, as the tension and compression principal stress are in the same direction. However, in inclination treatment, the location of tension principal stress differed, suggesting that new bone formation would be slower than bone resorption.

**Figure 18 materials-15-03118-f018:**
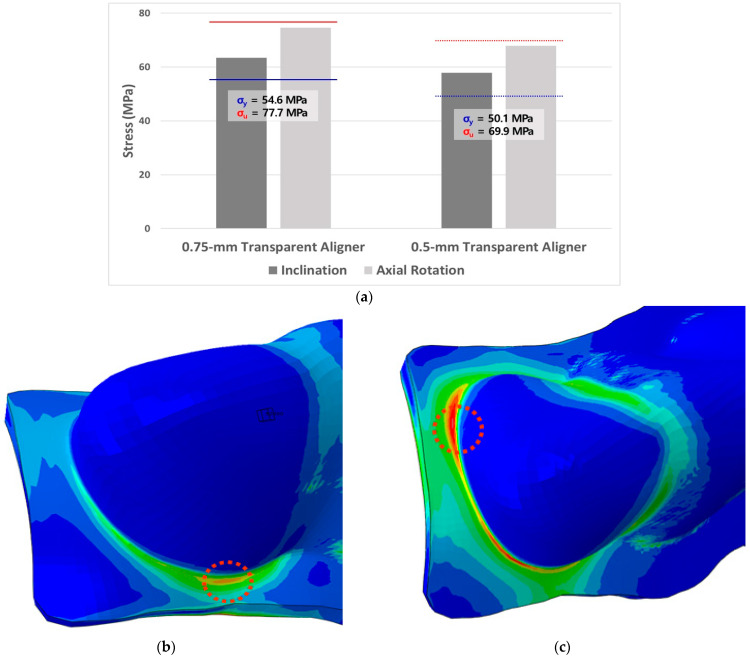
Peak von Mises stress (PVMS) of transparent aligners. (**a**), Both the 0.75-mm and 0.5-mm transparent aligners showed that PVMS was higher than the yield stress of the material after orthodontic treatment. Thus, restoring the original shape would be difficult; (**b**), In inclination treatment, both transparent aligners had the highest PVMS in the buccal direction in the area where the crown and gingiva of the central incisor were in contact; (**c**), In axial rotation correction, the highest PVMS was expected at the lingual-mesial point, suggesting that deformation and breakage of the area may be high after orthodontic treatment.

**Table 1 materials-15-03118-t001:** Linear Material Properties of the Finite Element Models.

Component		Elastic Modulus (MPa)	Poisson’s Ratio	Reference
Alveolar Bone	Cortical Bone	13,700	0.3	[17,21]
	Cancellous Bone	1370	0.3	[17,21]
Teeth		19,613	0.15	[17,21]
Gingiva		2.8	0.4	[17,21]
Fixed Appliance (bracket and arch-wire)		200,000	0.3	Provided by the manufacturer

## Data Availability

The data presented in this study are available on request from the corresponding author (S.-J.L.).

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
