# Peer review of "Biomechanical Efficacy and Effectiveness of Orthodontic Treatment with Transparent Aligners in Mild Crowding Dentition—A Finite Element Analysis"

_materials, 2022, doi:10.3390/ma15093118_

Round 1
Reviewer 1 Report
This is an interesting finite element study that compares the efficacy and effectiveness of orthodontic treatment with transparent aligners and of fixed appliances in simulated clinical orthodontic treatment conditions using orthodontic finite element (FE) models.
Although well thought out and a thorough study, a few issues must be addressed. Please see the enclosed pdf for details.

Reviewer 2 Report
Dear Seo et al.,
The manuscript “Biomechanical Efficacy and Effectiveness of Orthodontic Treatment with Transparent Aligners in Mild Crowding Dentition – A Finite Element Analysis” (materials-1701525) by Seo et al. conducted a FE analysis of orthodontic treatment using fixed appliances and transparent aligners on a tooth-alveolar bone FE model that consisted of alveolar bone, PDLs, and multiple teeth. The topic is interesting, but I think this article should reconsider after proper changes in major revision for publication in Materials. Some of my specific comments are:
- Describe the novelty of the article made by the author? From the results of my evaluation, it seems that many similar published works adequately explain what you have raised in the current manuscript related to finite element study of Orthodontic with aligners as the best reviewer knowledge in this research area. If there is something others really new in this manuscript, please highlight it more clearly in the introduction section (line 37-107).
- The state of the art and the significance of the present study are not clearly present, the authors should highlight it more advanced in the introduction section (line 37-107).
- In line 57-58, the authors stated that “To overcome these limitations, a removable appliance has 57 been developed. Align Technology (U.S) developed Invisalign, a transparent aligner 58 made of polymer materials”. There is only one possible solution to overcomes the problems? It must be use Align Technology (U.S) developed Invisalign, which no others product available in the market?
- Since the present study conduct finite element study of the medical implant, I would encourage and advise the authors to adopt some of the specific additional related references published in MDPI as follow:
-
- Tresca Stress Simulation of Metal-on-Metal Total Hip Arthroplasty during Normal Walking Activity. Materials (Basel). 2021, 14, 7554. https://doi.org/10.3390/ma14247554
- Mesh sensitivity study needs to be conducted to make sure the results accuracy of the present finite element model with certain number of elements. Is the presents study has been performing a mesh sensitivity study? If yes, please provide it in the present manuscript,
- Figure 5 (line 178-179), the authors state the figure described as “Orthodontic treatment finite element models”. However, the present illustration does not show elements on its model. Its is seems an Orthodontic 3D geometry model, not finite element model. If is is right, please revise the figure description for figure 5, also add the presentation of its finite element model.
- The authors need explain the element type used for generating finite element model.
- The meshing strategy for finite element model is missing in the present manuscript.
- For finite element model verification in the present study, why the authors use loads comparison in Figure 13 (line 438-440)? It is crucial to verify other results rather than only loading.
- The author must provide a detailed specification and use condition more detail regarding all tools used in the research carried out in the Materials and Methods section (line 108-370) so that the reader can estimate the accuracy and differences in the results that the authors describe due to the use of different tools in future studies.
- In the Results section (line 371-465), authors are advised to compare the results they obtain with previous similar/identical studies if it is possible.
- The conclusion (line 550-560) of the present manuscript is not solid. Further elaboration is needed.
- Further research needs to be explained in the conclusion section (line 550-560).
- To improve the quality of English used in this manuscript and make sure English language, grammar, punctuation, spelling, and overall style are correct, further proofreading is needed. As an alternative, the authors can use the MDPI English proofreading service for this issue.
I am pleased to have been able to review the author's present manuscript. Hopefully, the author can revise the current manuscript as well as possible so that it becomes even better. Good luck for the author's work and effort.
Best regards,
The Reviewer
Round 2
Reviewer 1 Report
The manuscript has been improved
Author Response
Thank you for your comments.

Reviewer 2 Report
Dear Seo et al.,
After carefully reading the author's revised manuscript entitled "Biomechanical Efficacy and Effectiveness of Orthodontic Treatment with Transparent Aligners in Mild Crowding Dentition – A Finite Element Analysis" (materials-1701525) by Seo et al., The authors have been made significant improvements in the revised manuscript. Also, all of the issues in my review report have been addressed precisely.
With my pleasure, I recommend the manuscript should be accepted for publication on Materials.
Best regards,
The Reviewer
Author Response
Thank you for your comments.
